# Towards Understanding Dual BN In Hybrid Adversarial Training

**Chenshuang Zhang**                                                                     *zcs15@kaist.ac.kr*
*KAIST*

**Chaoning Zhang** *                                                          *chaoningzhang1990@gmail.com*
*Kyung Hee University*

**Kang Zhang**                                                                 *zhangkang@kaist.ac.kr*
*KAIST*

**Axi Niu**                                                                    *nax@mail.nwpu.edu.cn*
*Northwestern Polytechnical University*

**Junmo Kim**                                                                 *junmo.kim@kaist.ac.kr*
*KAIST*

**In So Kweon**                                                               *iskweon77@kaist.ac.kr*
*KAIST*

**Reviewed on OpenReview:** *https://openreview.net/forum?id=bQKHMSE4SH*

## Abstract

There is a growing concern about applying batch normalization (BN) in adversarial training (AT), especially when the model is trained on both *adversarial* samples and *clean* samples (termed Hybrid-AT). With the assumption that *adversarial* and *clean* samples are from two different domains, a common practice in prior works is to adopt Dual BN, where $BN_{adv}$ and $BN_{clean}$ are used for adversarial and clean branches, respectively. A popular belief for motivating Dual BN is that estimating normalization statistics of this mixture distribution is challenging and thus disentangling it for normalization achieves stronger robustness. In contrast to this belief, we reveal that disentangling statistics plays a less role than disentangling affine parameters in model training. This finding aligns with prior work (Rebuffi et al., 2023), and we build upon their research for further investigations. We demonstrate that the domain gap between adversarial and clean samples is not very large, which is counter-intuitive considering the significant influence of adversarial perturbation on the model accuracy. We further propose a two-task hypothesis which serves as the empirical foundation and a unified framework for Hybrid-AT improvement. We also investigate Dual BN in test-time and reveal that affine parameters characterize the robustness during inference. Overall, our work sheds new light on understanding the mechanism of Dual BN in Hybrid-AT and its underlying justification.

## 1 Introduction

Adversarial training (AT) (Ganin et al., 2016; Madry et al., 2018; Shafahi et al., 2019; Andriushchenko & Flammarion, 2020; Bai et al., 2021) that optimizes the model on adversarial examples is a time-tested and effective technique for improving robustness against adversarial attack (Qiu et al., 2019; Xu & Yang,

---

*Corresponding author

2020; Dong et al., 2018; Zhang et al., 2021b). Beyond classical AT (also termed Madry-AT) (Madry et al., 2018), a common AT setup is to train the model on both *adversarial* samples and *clean* samples (termed Hybrid-AT) (Goodfellow et al., 2015; Kannan et al., 2018; Xie & Yuille, 2020; Xie et al., 2020a). Batch normalization (BN) (Ioffe & Szegedy, 2015; Santurkar et al., 2018; Bjorck et al., 2018; Li et al., 2017) has become a de facto standard component in modern deep neural networks (DNNs) (He et al., 2016; Huang et al., 2017; Zhang et al., 2019a; 2021a), however, there is a notable concern regarding how to use BN in the Hybrid-AT setup. This concern mainly stems from Xie & Yuille (2020); Xie et al. (2020a), which claims the adversarial and clean samples are from two different domains, and thus a separate BN should be used for each domain. This technique applying different BN for different domains has been adopted in multiple works with different names, e.g., Dual BN (Jiang et al., 2020; Wang et al., 2020; 2021) and mixture BN (Xie & Yuille, 2020). With different names, however, they refer to the same practice of adopting $BN_{adv}$ and $BN_{clean}$ for adversarial and clean samples, respectively. To avoid confusion, we use Dual BN for the remainder of this work.

Despite the increasing popularity of Dual BN, the mechanism of how Dual BN helps Hybrid-AT remains not fully clear. Toward a better understanding of this mechanism, we revisit a long-held belief in Xie & Yuille (2020); Xie et al. (2020a). Specifically, it justifies the necessity of Dual BN in Hybrid-AT with the following claim (quoted from the abstract of Xie & Yuille (2020)):

*"Estimating normalization statistics of the mixture distribution is challenging"* and *"disentangling the mixture distribution for normalization, i.e., applying separate BNs to clean and adversarial images for statistics estimation, achieves much stronger robustness."*

The above claim (Xie & Yuille, 2020) emphasizes the necessity of disentangling the normalization statistics (NS) in Hybrid-AT. The underlying motivation for the above claim is that BN statistics calculated on the clean domain are incompatible with training the model on the adversarial domain, and vice versa. Therefore, Hybrid-AT with a single BN suffers from such incompatibility with BN statistics calculated from the mixed distribution, while Dual BN can avoid the incompatibility through training the clean and adversarial samples with two BN branches separately. As a preliminary investigation, our work experiments with a new variant of AT with Cross-BN, namely training the adversarial samples with $BN_{clean}$ and vice versa. Interestingly, we find that using BN from another domain only has limited influence on the performance. This observation inspires us to have a closer look at how Dual BN works in Hybrid-AT. Through untwining normalization statistics (NS) and affine parameters (AP) in Dual BN to include one effect while excluding the other, we demonstrate that two AP sets can achieve comparable performance to the original Dual BN, which is consistent with the finding in Rebuffi et al. (2023). We also reveal that disentangled NS can achieve similar performance to Dual BN under certain conditions like small perturbations ($\epsilon = 8/255$). These findings refute the prior claim emphasizing the role of disentangled NS in Dual BN (Xie & Yuille, 2020; Xie et al., 2020a), and also inspires us to investigate whether the motivation for Dual BN holds, i.e., the two-domain hypothesis in Xie & Yuille (2020); Xie et al. (2020a).

As the motivation for adopting Dual BN, the two-domain hypothesis assumes that *"clean images and adversarial images are drawn from two different domains"* (quoted from Xie & Yuille (2020)). This hypothesis is verified in Xie & Yuille (2020) mainly by the visualization of NS, which highlights a large adversarial-clean domain gap. However, we point out that their visualization has a hidden flaw, which makes their claim regarding the domain gap between adversarial and clean samples deserve a closer look. Specifically, the visualization in Xie & Yuille (2020) ignores the influence of different AP when calculating NS. After fixing this hidden flaw, we demonstrate that the adversarial-clean domain gap is not as large as claimed in prior work. Interestingly, under the same perturbation/noise magnitude, we show that there is no significant difference between adversarial-clean domain gap and noisy-clean counterpart.

Inspired by the above findings, we propose a two-task hypothesis to replace the two-domain hypothesis in Xie & Yuille (2020); Xie et al. (2020a) for justification on how Dual BN works in Hybrid-AT. Specifically, we claim that there are two tasks in Hybrid-AT: one task for clean accuracy and the other for robustness. Our two-task hypothesis offers empirical foundations and a unified framework for Hybrid-AT improvements, which generalizes Hybrid-AT with Dual BN to various model designs, including the adapter method in Rebuffi et al. (2023) and Trades-AT (Zhang et al., 2019b). In addition to exploring BN for training Hybrid-

AT models, our study delves into Dual BN at test time, uncovering that affine parameters characterize robustness during inference.

We summarize our main contributions as follows.

- Our work thoroughly investigates how disentangled normalization statistics (NS) and affine parameters (AP) in Dual BN impact the training Hybrid-AT models, leading to a comprehensive and solid refutation of prior claims about the significance of NS.

- Our work investigates the adversarial-clean domain gap. We point out a hidden flaw of NS visualization in prior work, and demonstrate the adversarial-clean domain gap is not as large as expected both visually and quantitatively.

- Our work proposes a two-task hypothesis as an empirical foundation and unified framework for enhancing Hybrid-AT, connecting diverse methods like Dual BN, Dual Linear, Adapters and Trades-AT. This hypothesis may bring new inspirations to Hybrid-AT improvements from a new perspective.

- Our study examines Dual BN at test time, exploring various NS and AP types with a pretrained model and revealing that AP determines robustness during inference.

## 2 Problem overview

### 2.1 Adversarial training

**Adversarial training.** Adversarial training (AT) (Ganin et al., 2016; Madry et al., 2018; Shafahi et al., 2019; Andriushchenko & Flammarion, 2020; Bai et al., 2021) has been the most powerful defense method against adversarial attacks, among which Madry-AT (Madry et al., 2018) is a typical method detailed as follows. Let's assume $\mathcal{D}$ is a data distribution with $(x, y)$ pairs and $f(\cdot, \theta)$ is a model parametrized by $\theta$. $l$ indicates cross-entropy loss in classification. Instead of directly feeding clean samples from $\mathcal{D}$ to minimize the risk of $\mathbb{E}_{(x,y)\sim\mathcal{D}}[l(f(x,\theta), y)]$, Madry et al. (2018) formulates a saddle problem for finding model parameter $\theta$ by optimizing the following adversarial risk:

$$\arg\min_{\theta} \mathbb{E}_{(x,y)\sim\mathcal{D}} \left[ \max_{\delta\in\mathbb{S}} l(f(x+\delta;\theta), y) \right] \tag{1}$$

where $\mathbb{S}$ denotes the allowed perturbation budget which is a typically $l_p$ norm-bounded $\epsilon$. We term the above adversarial training framework as Classical-AT. It adopts a two-step training procedure (inner maximization + outer minimization), and trains the robust model with only adversarial samples. Following the same procedure, Xie & Yuille (2020); Xie et al. (2020a) propose to train the robust model with both clean and adversarial samples, termed as **Hybrid-AT**. The loss of Hybrid-AT is defined as follows:

$$\mathcal{L}_{Hybrid} = \alpha l(f(x;\theta), y) + (1-\alpha)l(f(x+\delta;\theta), y) \tag{2}$$

where $x$ and $x+\delta$ indicate clean and adversarial samples, respectively. $\alpha$ is a hyper-parameter for balancing the clean and adversarial branches, is set to 0.5 in this work following Goodfellow et al. (2015); Xie & Yuille (2020).

### 2.2 Batch normalization in AT

**Batch normalization (BN).** We briefly summarize how BN works in modern networks. For a certain layer in the DNN, we denote the feature layers of a mini-batch in the DNN as $\mathcal{B} = \{x^1, ..., x^m\}$. The feature layers are normalized by mean $\mu$ and standard deviation $\sigma$ as:

$$\hat{x}^i = \frac{x^i - \mu}{\sigma} \cdot \gamma + \beta \tag{3}$$

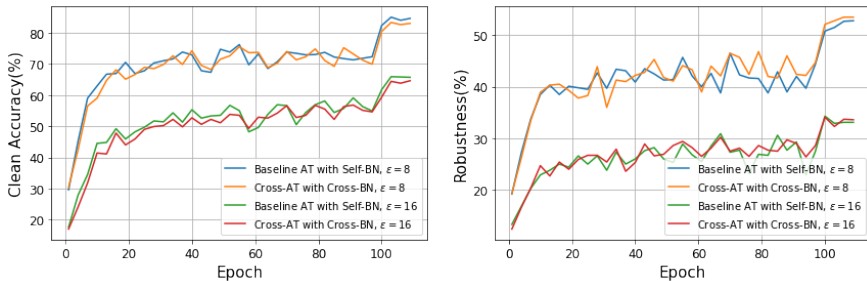

Figure 1: Clean accuracy and robustness (PGD10 Accuracy) of Cross-AT during training. In Cross-AT, the *adversarial* samples are normalized by the BN statistics calculated by *clean* samples. Interestingly, Cross-AT yield comparable robustness to original Self-BN(BN$_{adv}$).

Table 1: Test accuracy of Cross-Hybrid-AT ($\epsilon = 16/255$). In Cross-Hybrid-AT, the adversarial branch is normalized by BN$_{clean}$, and the clean branch is normalized by BN$_{adv}$. Experimental results show that Cross-Hybrid-AT achieves comparable results to Hybrid-AT with vanilla Dual BN.

| Model | Training | Test | Clean | PGD-10 | AA |
|---|---|---|---|---|---|
| Hybrid-AT | Dual BN | BN$_{adv}$ | 61.84 | 31.67 | 22.51 |
| Cross-Hybrid-AT | Dual BN | BN$_{clean}$ | 59.56 | 31.25 | 22.40 |
| Hybrid-AT | Single BN | | 93.70 | 29.86 | 0.48 |

where $\gamma$ and $\beta$ indicate the weight and bias in BN, respectively. To be clear, we refer $\mu$ and $\sigma$ as normalization statistics (NS), $\gamma$ and $\beta$ as affine parameters (AP). During training, NS is calculated on the current mini-batch statistics for the update of model weights. Meanwhile, a running average of NS is recorded in the whole training process, which is applied for inference after training ends.

**Dual BN in AT.** There is an increasing interest in investigating BN in the context of adversarial robustness (Awais et al., 2020; Cheng et al., 2020; Nandy et al., 2021; Sitawarin et al.; Gong et al., 2022). This work focuses on Hybrid-AT with Dual BN (Xie & Yuille, 2020; Xie et al., 2020a) which applies BN$_{clean}$ and BN$_{adv}$ to clean branch and adversarial branch, respectively.

### 2.3 Experimental setups

Pang et al. (2020) demonstrates that AT's basic training settings significantly impact model performance and recommended specific parameters for a fair comparison of AT methods. If not specified, we adhere to the suggested settings in Pang et al. (2020).

**Experimental setups.** In this work, we perform experiments on CIFAR10 (Krizhevsky et al., 2009; Andriushchenko & Flammarion, 2020; Zhang et al., 2022) with ResNet18 (Andriushchenko & Flammarion, 2020; Targ et al., 2016; Wu et al., 2019; Li et al., 2016; Zhang et al., 2022). Specifically, we train the model for 110 epochs. The learning rate is set to 0.1 and decays by a factor of 0.1 at the epoch 100 and 105. We adopt an SGD optimizer with weight decay $5 \times 10^{-4}$. For generating adversarial examples during training, we use $\ell_\infty$ PGD attack with 10 iterations and step size $\alpha = 2/255$. For the perturbation constraint, $\epsilon$ is set to $\ell_\infty$ 8/255 (Pang et al., 2020) or 16/255 (Xie & Yuille, 2020). Following Pang et al. (2020), we evaluate the model robustness under PGD-10 attack (PGD attack with 10 steps) and AutoAttack (AA) (Croce & Hein, 2020).

## 3 On the BN induced misalignment

In Hybrid-AT, the model is trained with two branches: a clean branch and an adversarial branch. These two branches share all model weights but are found to require independent BN modules, i.e., Dual BN (Xie & Yuille, 2020; Xie et al., 2020a). At test time, only a single branch can be used by choosing either BN$_{adv}$

Table 3: Test accuracy (%) of untwining NS and AP in Dual BN. For NSs, 1 indicates mixed distribution and 2 indicates disentangled distribution for normalization. For APs, 1 indicates a single set and 2 indicates double sets of APs. The subscripts of $AP_{adv}$ and $AP_{clean}$ indicate the input data type used during training. Setup1 with two sets of APs achieves comparable results with Dual BN.

| Setups | NS | AP | $\epsilon = 8/255$ | | | $\epsilon = 16/255$ | | |
|---|---|---|---|---|---|---|---|---|
| | | | Clean | PGD-10 | AA | Clean | PGD-10 | AA |
| Single BN | 1 | 1 | 88.06 | 49.75 | 7.03 | 93.70 | 29.86 | 0.48 |
| Dual BN ($BN_{adv}$) | 2 | 2 | 82.77 | 51.33 | 46.19 | 61.84 | 31.67 | 23.14 |
| Dual BN ($BN_{clean}$) | 2 | 2 | 94.91 | 0.32 | 0.10 | 94.18 | 0.00 | 0.00 |
| Setup1 ($AP_{adv}$) | 1 | 2 | 81.86 | 50.99 | 44.63 | 60.02 | 30.89 | 23.43 |
| Setup1 ($AP_{clean}$) | 1 | 2 | 94.74 | 0.10 | 0.04 | 94.30 | 0.00 | 0.00 |
| Setup2 ($NS_{adv}$) | 2 | 1 | 85.49 | 49.39 | 42.96 | 55.91 | 21.92 | 10.64 |
| Setup2 ($NS_{clean}$) | 2 | 1 | 89.22 | 49.48 | 42.95 | 86.35 | 1.08 | 0.00 |

or $BN_{clean}$. The adversarial branch (with $BN_{adv}$) is adopted in Xie & Yuille (2020) for prioritizing high model robustness, while $BN_{clean}$ is adopted in Xie et al. (2020a) for only considering clean accuracy.

However, swapping the BN during inference, i.e., adopting $BN_{clean}$ for robustness and $BN_{adv}$ for clean accuracy, leads to a significant performance drop. As shown in Table 2, $BN_{clean}$ leads to almost zero robustness during inference. This interesting phenomenon inspires us to investigate the following question: *will $BN_{clean}$ achieve robustness if it is trained with the adversarial branch, and vice versa?* To facilitate discussion of the above misalignment, we introduce a new term **Cross-BN** which refers to adopting $BN_{clean}$ for the adversarial branch or $BN_{adv}$ for the clean branch. With a similar terminology rule, $BN_{clean}$ for the clean branch or $BN_{adv}$ for the adversarial branch is termed as **Self-BN**.

Table 2: Test accuracy (%) of Hybrid-AT with Dual BN. $BN_{clean}$ leads to almost zero robustness under both perturbation budgets ($\epsilon$): 8/255 and 16/255.

| $\epsilon$ | Setups | Clean | PGD-10 | AA |
|---|---|---|---|---|
| 8/255 | Dual BN ($BN_{adv}$) | 82.77 | 51.33 | 46.19 |
| | Dual BN ($BN_{clean}$) | 94.91 | 0.32 | 0.10 |
| 16/255 | Dual BN ($BN_{adv}$) | 61.84 | 31.67 | 23.14 |
| | Dual BN ($BN_{clean}$) | 94.18 | 0.00 | 0.00 |

**Cross-AT: a preliminary investigation.** Before investigating Hybrid-AT with Cross-BN, we first investigate a setting where *only* adversarial samples are used for model training. Note that it is adversarial branch, and the baseline model with a Self-BN adopts $BN_{adv}$. Cross-AT is conducted by replacing the default $BN_{adv}$ with a Cross-BN, i.e., $BN_{clean}$ (see Figure 2). Specifically, the adversarial samples are normalized by the BN statistics calculated by clean samples. It should be noted that in Cross-AT, the clean samples are used only for forward propagation to get the BN statistics, and the model weights are updated only by the adversarial branch. Interestingly, although the adversarial branch is normalized by BN*clean*, Figure 1 shows that Cross-AT achieves comparable performance as the baseline model with Self-BN($BN_{adv}$).

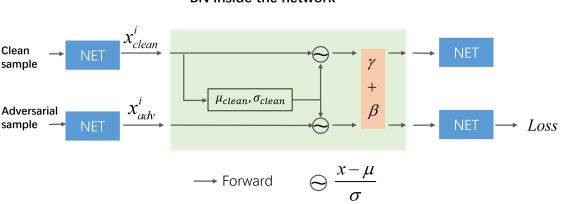

Figure 2: Cross-AT: Replacing $BN_{adv}$ with $BN_{clean}$ in the adversarial branch. The adversarial samples are normalized by the BN statistics calculated by clean samples.

**Cross-Hybrid-AT: Hybrid-AT with Cross-BN.** Here, for the Dual BN in Hybrid-AT, we replace the default Self-BN with Cross-BN and term it Cross-Hybrid-AT. In Cross-Hybrid-AT, the adversarial branch is normalized by $BN_{clean}$, and the clean branch is normalized by $BN_{adv}$. As shown in Table 1, $BN_{clean}$ in Cross-Hybrid-AT achieves comparable results to $BN_{adv}$ in Hybrid-AT. The finding in Cross-Hybrid-AT is consistent with that in Cross-AT, which indicates that Cross-BN achieves comparable results to Self-BN.

**Implication of the above results.** As discussed above, training the model with Cross-BN leads to a comparable performance as with Self-BN in Hybrid-AT. However, this finding appears counter-intuitive considering the results of Hybrid-AT with Single BN. As shown in Table 1, Single BN leads to almost zero robustness (0.48%) under AA attack. Note that a single BN is calculated by a mixture of clean and adversarial samples. If calculating BN statistics on either clean examples or adversarial examples can lead to high robustness, how come training on BN calculated on hybrid samples leads to an AA robustness close

to zero? This finding also conflicts with the claims in Xie & Yuille (2020); Xie et al. (2020a) that emphasize the importance of NS, and inspire us to investigate how Dual BN works in Hybrid-AT.

## 4    Understanding how Dual BN works in training of Hybrid-AT

The failure of prior claims Xie & Yuille (2020); Xie et al. (2020a) to explain our observations in Section 3 inspires us to investigate how Dual BN works in Hybrid-AT. On top of the single BN as a default case, Dual BN introduces an auxiliary BN component and causes two changes: (i) disentangling the mixture distribution for normalization statistics (NS) and (ii) introducing two sets of affine parameters (AP). To fully understand Dual BN in Hybrid-AT, we delve into its mechanisms.

**Untwining NS and AP in Dual BN.** As discussed above, compared with Hybrid-AT with Single-BN, Dual BN brings two effects: disentangled NSs and two sets of APs. To determine the influence of each effect on the model performance, we design two setups of experiments to include only one effect while excluding the other. In Setup1, we only include the effect of two sets of APs, by applying two different sets of APs ($\beta_{adv}/\gamma_{adv}$ and $\beta_{clean}/\gamma_{clean}$) in the adversarial and clean branches while using the default mixture distribution for normalization. In Setup2, we only include the effect of two sets of NSs by disentangling this mixture distribution with two different sets of NSs while making $\text{BN}_{clean}$ and $\text{BN}_{adv}$ share the same set of APs. The above setups of BNs are summarized in Figure 3 and we discuss the experimental results in Table 3 as follows.

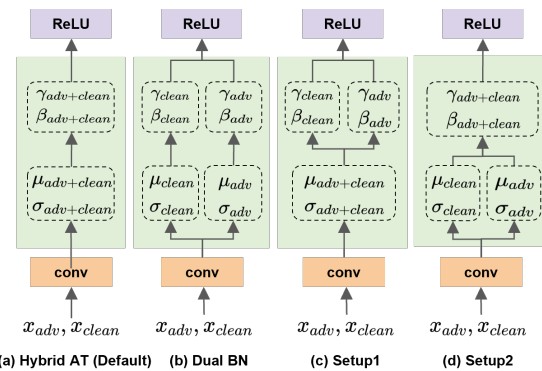

Figure 3: Illustration of different BN setups for untwining NS and AP in Dual BN of Hybrid-AT.

**Role of disentangled NS and AP.** As shown in Table 3, Dual BN (with $\text{BN}_{adv}$ during inference) brings significant robustness improvement over the Single BN baseline, which is consistent with findings in Xie & Yuille (2020). Interestingly, under the attack of PGD-10, their robustness gap is not significant, however, under AA, the Single BN achieves very low robustness (7.03% and 0.48% for $\epsilon = 8/255$ and $16/255$, respectively). Moreover, Setup1 ($\text{AP}_{adv}$) achieves comparable robustness as that of Dual BN ($\text{BN}_{adv}$) for $\epsilon = 8/255$ and $16/255$, suggesting two sets of APs alone achieve similar performance as Dual BN for yielding higher robustness ($\text{AP}_{adv}$) than single BN setting. The effect of two sets of NSs is more nuanced: for a small perturbation $\epsilon = 8/255$, disentangling mixture distribution is beneficial for boosting the robustness under strong AA; for a large perturbation $\epsilon = 16/255$, this benefit is less significant. This can be explained by the fact that training under $\epsilon = 16/255$ is much harder than $\epsilon = 8/255$.

**Conclusions.** Overall, we have two conclusions. First, two sets of AP achieve comparable performance to Dual BN, aligning with the findings in Rebuffi et al. (2023). Moreover, our research extends beyond Rebuffi et al. (2023) by not only disentangling AP but also exploring the disentanglement of NS for a more thorough examination of Dual BN. Although not as effective as disentangling AP, disentangling NS can also achieve comparable robustness to Dual BN under certain conditions even against the strong AutoAttack, such as the small perturbation($\epsilon = 8/255$). However, the benefit of disentangling NS narrows significantly for large perturbation.

## 5    On the domain gap between clean and adversarial samples

A model trained on a source domain performs poorly on a new target domain when there is a domain shift (Daumé III, 2007; Sun et al., 2017). With BN as the target, it is common in the literature (Li et al., 2017; Benz et al., 2021a; Schneider et al., 2020; Xie & Yuille, 2020; Xie et al., 2020a) to indicate the domain gap by the difference of NS between two domains. In adversarial machine learning, prior work (Xie & Yuille, 2020; Xie et al., 2020a; Jiang et al., 2020) perceive the adversarial domain as a new domain. Specifically, Xie & Yuille (2020) highlights the adv-clean domain gap by visualizing the difference of NS in $\text{BN}_{adv}$ and $\text{BN}_{clean}$

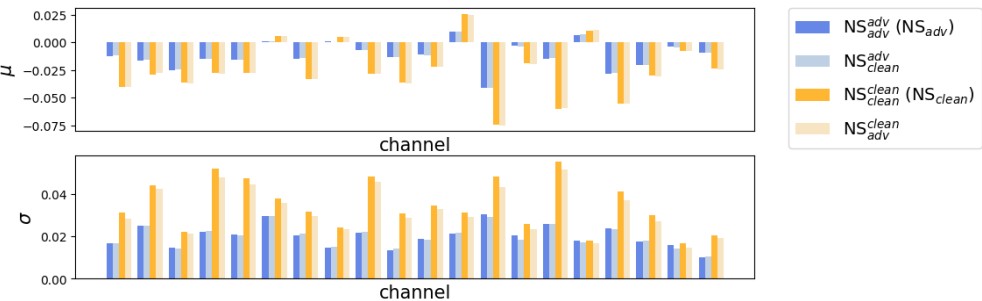

Figure 4: Visualization of normalization statistics (NS) by randomly choosing 20 channels and displaying the NS calculated with different APs. The superscript and subscript of NS refer to the AP and input images when calculating NS, respectively. For example, $\text{NS}_{clean}^{adv}$ is computed on clean samples with $\text{AP}_{adv}$. NSs calculated by the same AP are close to each other, such as $\text{NS}_{adv}^{adv}$ and $\text{NS}_{clean}^{adv}$ calculated by $\text{AP}_{adv}$, so is similar $\text{NS}_{clean}^{clean}$ and $\text{NS}_{adv}^{clean}$ calculated by $\text{AP}_{clean}$.

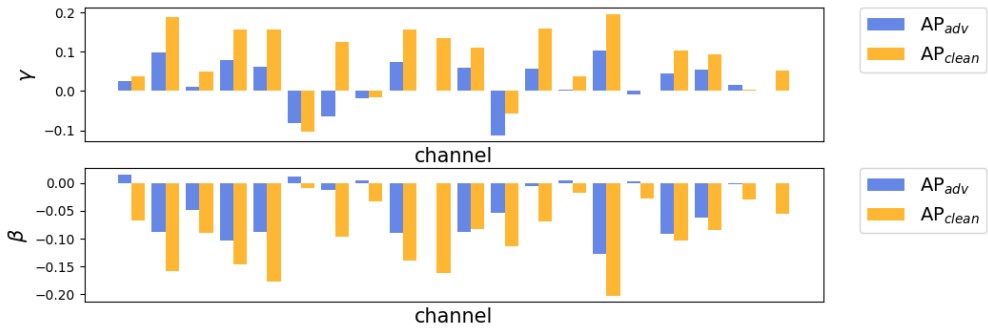

Figure 5: Visualization of affine parameters (AP). Randomly chose 20 channels for visualizing $\text{AP}_{clean}$ and $\text{AP}_{adv}$. There exists a gap between $\text{AP}_{clean}$ and $\text{AP}_{adv}$.

(see Figure 5 of (Xie & Yuille, 2020)). The large domain gap visualized in (Xie & Yuille, 2020) is somewhat in conflict with the finding in Section 4 that disentangling NS plays a less role for high performance. We further investigate the adv-clean domain gap for a comprehensive understanding.

## 5.1 A hidden flaw leads to a misleading visualization

With our analysis in Section. 4, we know that the AP in $\text{BN}_{clean}$ and $\text{BN}_{adv}$ are different. The clean branch and adversarial branch still have different weights, i.e., AP, even though the same set of convolutional filters are shared. Therefore, the NS difference between $\text{BN}_{clean}$ and $\text{BN}_{adv}$ is characterized by two factors: (a) AP inconsistency and (b) different domain inputs. We discuss the influence of these two factors on the NS difference as follows.

**Re-calibrated NS for disentangled analysis.** In the default setup of Dual BN, $\text{NS}_{clean}$ is calculated on clean samples with $\text{AP}_{clean}$, while $\text{NS}_{adv}$ is calculated on adversarial samples with $\text{AP}_{adv}$. In order to analyze the influence of different AP and domain inputs on the NS, we additionally calculate the NS on clean samples with $\text{AP}_{adv}$ (denoted as $\text{NS}_{clean}^{adv}$) and calculate the NS on adversarial samples with $\text{AP}_{clean}$ (denoted as $\text{NS}_{adv}^{clean}$). These two NS are termed **re-calibrated NS** since the AP and inputs are from different branches. Following $\text{NS}_{clean}^{clean}$ and $\text{NS}_{clean}^{adv}$ to indicate AP choice with the superscript and indicate sample choice with the subscript, we can also denote vanilla $\text{NS}_{clean}$ as $\text{NS}_{clean}^{clean}$ and denote $\text{NS}_{adv}$ as $\text{NS}_{adv}^{adv}$. Both $\text{NS}_{clean}^{clean}$ and $\text{NS}_{adv}^{adv}$ are termed as **vanilla NS** for differentiation. Details of obtaining various NS is reported in Section A.1 of the appendix.

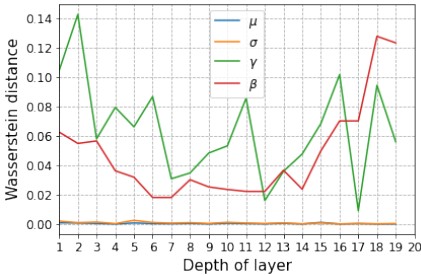

Figure 6: Layer-wise discrepancy visualization. For all layers, there exists a large distance (higher than zero) between $\text{AP}_{clean}$ and $\text{AP}_{adv}$, see $\gamma$ and $\beta$ in the figure. However, with the same $\text{AP}_{adv}$, the gap between $\text{NS}^{adv}_{adv}$ and $\text{NS}^{adv}_{clean}$ stays almost zero in all layers, see $\mu$ and $\sigma$ in the figure.

**A hidden flaw of NS visualization in Xie & Yuille (2020).** To exclude the influence of AP inconsistency, we intend to compare NS between clean and adversarial samples with the same AP (the superscript in NS). In other words, the domain gap is characterized by the difference between $\text{NS}^{clean}_{clean}$ and $\text{NS}^{clean}_{adv}$ or that between $\text{NS}^{adv}_{adv}$ and $\text{NS}^{adv}_{clean}$. Following the procedures in Xie & Yuille (2020), we plot different types of NS in Figure 4 by randomly sampling 20 channels of the second BN layer in the first residual block. Fig. 4 shows that there exists a gap between $\text{NS}^{clean}_{clean}$ and $\text{NS}^{adv}_{adv}$, which is consistent with the findings in Xie & Yuille (2020). Moreover, there are two other observations from Figure 4. First, if we fix the input samples and calculate NS with different AP, there exists a large gap, i.e., the gap between $\text{NS}^{adv}_{clean}$ and $\text{NS}^{clean}_{clean}$, as well as the gap between $\text{NS}^{adv}_{adv}$ and $\text{NS}^{clean}_{adv}$. Second, those NSs with the same APs are very close to each other: $\text{NS}^{adv}_{adv}$ and $\text{NS}^{adv}_{clean}$ are very similar to each other, and the same applies to $\text{NS}^{clean}_{adv}$ and $\text{NS}^{clean}_{clean}$. We report the visualization results of AP in Figure 5 for comparison, which shows a significant gap between $\text{AP}_{clean}$ and $\text{AP}_{adv}$.

**Conclusions.** We point out a flaw in Xie & Yuille (2020) that the large adv-clean gap visualized in Xie & Yuille (2020) is caused by the AP consistency. When adopting the same AP, the adv-clean domain gap significantly narrows. Our investigations suggest that the visualization and conclusions in Xie & Yuille (2020) might convey a misleading message. Our findings update the understanding on the adv-clean domain gap.

## 5.2 Adv-clean domain gap is not as large as expected

**Quantitative measurement of domain gap.** Figure 4 investigates the adv-clean domain gap qualitatively. For a quantitative comparison, we measure the Wasserstein distance between clean and adversarial branches in different layers in Figure 6. As shown in Figure 6, the Wasserstein distance of NS between clean and adversarial branches is much smaller than the difference of AP for a certain layer. This finding is consistent with that in Figure 4 and Figure 5.

**Adv-clean versus noisy-clean domain gap.** As suggested in Benz et al. (2021a); Schneider et al. (2020), noisy samples (images corrupted by random noise) can be seen as a domain different from clean samples. Adversarial perturbation is a *worst-case* noise for attacking the

Table 4: Test accuracy (%) under random noise and adversarial perturbation during inference.

| Noise/perturbation Size | 0 | 8/255 | 16/255 |
|---|---|---|---|
| Random noise | 94.0 | 92.7 | 86.6 |
| Adversarial perturbation | 94.0 | 0.00 | 0.00 |

model. Taking a ResNet18 model trained on clean samples for example, we report the performance under adversarial perturbation and random noise (with the same magnitude) in Table 4. As expected, the model accuracy drops to zero with adversarial perturbation. Under random noise of the same magnitude, we find that the model performance only drops by a small margin. Given that the influence of adversarial perturbation on the model performance is significantly larger than that of random noise, it might be tempting to believe that the adversarial-clean domain gap is much larger than noisy-clean domain gap.

With Wasserstein distance of NS between different domains as the metric, we compare the adversarial-clean domain gap with noisy-clean counterpart on the above ResNet18 model trained on clean samples, as shown in Figure 7. The perturbation and noise magnitude are set to 16/255. Interestingly, we observe that there is no significant difference between the adversarial-clean domain gap and noisy-clean counterpart.

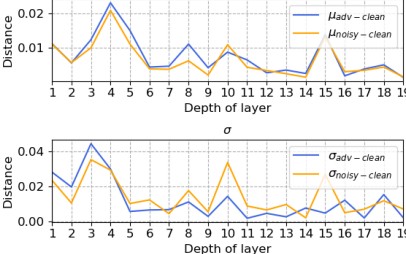

Figure 7: Visualization of adversarial-clean domain gap and noisy-clean domain gap (perturbation/noise magnitude is set to 16/255).

**Conclusions.** Based on the visualization and quantitative results, we reveal that the adversarial-clean domain gap is not as large as many might expect, considering the strong performance drop caused by adversarial perturbation.

### 5.3 Interpreting Hybrid-AT from a two-task perspective

**New empirical foundation for Hybrid-AT improvement.** Prior work (Xie & Yuille, 2020) takes the two-domain hypothesis as the empirical foundation for applying Dual BN in Hybrid-AT. Our investigations in Section 5.1 and Section 5.2 identify a flaw in the two-domain hypothesis, and point out that the adv-clean domain gap is not as large as expected. These findings suggest discarding the two-domain hypothesis in Hybrid-AT improvement.

Table 5: Test accuracy of Hybrid-AT with Dual linear, $\epsilon = 16/255$. Hybrid-AT with Dual Linear achieves a similar trend and comparable results with Dual BN.

| Setups | Branch | Clean | PGD10 | AA |
|---|---|---|---|---|
| Dual BN | $BN_{adv}$ | 61.84 | 31.67 | 22.51 |
| | $BN_{clean}$ | 94.18 | 0.00 | 0.00 |
| Dual Linear | $Linear_{adv}$ | 60.72 | 28.84 | 16.50 |
| | $Linear_{clean}$ | 91.43 | 2.21 | 1.30 |

Instead, we propose a new empirical foundation for Hybrid-AT improvement: the two-task hypothesis. Intuitively, with the two branches in Hybrid-AT, the model weights are trained for two tasks: one for clean accuracy and the other for robustness. A common approach for handling two tasks with a shared backbone is to make the top layers unshared. Here, we experiment with a shared encoder of single BN but with dual linear classifiers. The results in Table 5 show that Dual Linear results in similar behavior as Dual BN, validating the two-task hypothesis. Rebuffi et al. (2023) proposes to use adapters for each type of input in Hybrid-AT model, which matches the classification performance of Dual BN with significantly fewer parameters. The success of adapters in Rebuffi et al. (2023) also validates our two-task hypothesis.

**A unified framework for Hybrid-AT improvements.** Our new perspective on Hybrid-AT enables alternative solutions to mitigate the two-task conflict without resorting to two sets of APs. Here, we experiment with including an additional regularization loss, which is introduced to minimize the gap between two tasks. As a concrete example, we add a KL loss on the basic loss of Hybrid-AT in Eq 2. The

Table 6: Test accuracy of Hybrid-AT (Single BN) with KL loss, $\epsilon = 16/255$.

| Setups | Clean | PGD10 | AA |
|---|---|---|---|
| Single BN | 93.70 | 29.86 | 0.48 |
| Single BN (with KL loss) | 68.86 | 33.61 | 23.60 |
| Dual BN ($BN_{adv}$) | 61.84 | 31.67 | 22.51 |
| Dual BN ($BN_{clean}$) | 94.18 | 0.00 | 0.00 |

extra loss is designed to explicitly minimize the discrepancy between the outputs of adversarial and clean branches. Interestingly, this simple change improves the AA result of Single BN significantly from 0.48% to 23.60%. Compared to $BN_{adv}$ (the default branch during inference), Hybrid-AT with KL loss achieves superior performance on both clean accuracy and robustness. Interestingly, the Hybrid-AT with KL loss reminds us of another AT framework termed Trades-AT (Zhang et al., 2019b), which is also trained on hybrid samples and has a KL loss. This might provide an explanation for the effectiveness of Trades-AT (Zhang et al., 2019b) by analyzing the KL term. Admittedly, KL loss on the output is just a naive attempt, but its promising result invites future works to explore other solutions.

**Conclusions.** Our investigations on the adv-clean domain gap suggest discarding the prior two-domain hypothesis. Instead, we propose a two-task hypothesis as the new empirical foundation for Hybrid-AT improvement. The two-task hypothesis also serves as a unified framework that collaborates different methods

Table 7: Evaluation results of various NS and AP pairs, $\epsilon = 16/255$. During inference, **re-calibrated** NS achieves comparable performance to the default setting.

|  | Setups | NS | AP | $\epsilon = 8/255$ | | $\epsilon = 16/255$ | |
|---|---|---|---|---|---|---|---|
|  |  |  |  | PGD10 | AA | PGD10 | AA |
| Default | $\text{BN}_{adv}$ | $\text{NS}_{adv}^{adv}$ | $\text{AP}_{adv}$ | 51.33 | 46.19 | 31.67 | 22.51 |
|  | $\text{BN}_{clean}$ | $\text{NS}_{clean}^{clean}$ | $\text{AP}_{clean}$ | 0.32 | 0.10 | 0.00 | 0.00 |
| Swap | Setup1 | $\text{NS}_{clean}^{clean}$ | $\text{AP}_{adv}$ | 17.1 | 9.16 | 10.02 | 9.80 |
|  | Setup2 | $\text{NS}_{adv}^{adv}$ | $\text{AP}_{clean}$ | 0.00 | 0.00 | 0.45 | 0.00 |
| Re-calibration | Setup3 | $\text{NS}_{clean}^{adv}$ | $\text{AP}_{adv}$ | 51.75 | 46.55 | 32.73 | 24.40 |
|  | Setup4 | $\text{NS}_{adv}^{clean}$ | $\text{AP}_{clean}$ | 0.00 | 0.00 | 0.00 | 0.00 |

together, such as Dual BN, adapters in Rebuffi et al. (2023) and Trades-AT (Zhang et al., 2019b) that seem to be unrelated at first sight.

### 5.4 AP characterizes robustness in test-time

Investigation on BN in test-time has been a widely discussed topic in other areas (Li et al., 2017; Benz et al., 2021a; Schneider et al., 2020). Different from Section 4 and Rebuffi et al. (2023) that investigates model training, we further investigate how Dual BN works in test-time. Specifically, we compare the robustness with various NS and AP combinations, as reported in Table 7. Details of obtaining various NS are reported in Section A.1 of the appendix.

**Robustness with various NS and AP pairs.** Table 7 shows that given $\text{AP}_{adv}$, re-calibrated $\text{NS}_{clean}^{adv}$ achieves a robustness of 51.75%, which is comparable to 51.33% with $\text{NS}_{adv}^{adv}$. Note that the only difference between $\text{NS}_{clean}^{adv}$ and $\text{NS}_{adv}^{adv}$ is that they are calculated by clean and adversarial samples, respectively. Moreover, given $\text{AP}_{clean}$, both $\text{NS}_{adv}^{clean}$ and $\text{NS}_{clean}^{clean}$ yield zero robustness. The results of swapping $\text{NS}_{clean}^{clean}$ and $\text{NS}_{adv}^{adv}$ when AP is fixed is also given in Table 7 for comparison. We find that directly replacing $\text{NS}_{adv}^{adv}$ with $\text{NS}_{clean}^{clean}$ in $\text{BN}_{adv}$ (Setup1) results in lower robustness (17.1%) than original $\text{BN}_{adv}$ (51.33%).

**Conclusions.** We conclude that AP characterizes the large robustness gap between $\text{BN}_{clean}$ and $\text{BN}_{adv}$ during inference. When AP is consistent for both NS computation and robustness evaluation, the robustness gap between the NS calculated on clean or adversarial samples is limited.

## 6 Related work

**Adversarial training.** Since the advent of Classical-AT (Madry et al., 2018) and Hybrid-AT (Xie & Yuille, 2020; Xie et al., 2020a), numerous works have attempted to improve AT from various perspectives. From the data perspective, Uesato et al. (2019); Carmon et al. (2019); Zhang et al. (2019c) have independently shown that unlabeled data can be used to improve the robustness. From the model perspective, AT often benefits from the increased model capacity of models (Uesato et al., 2019; Xie & Yuille, 2020). Xie et al. (2020b); Pang et al. (2020); Gowal et al. (2020) have investigated the influence and suggested that a smooth activation function, like parametric softplus, is often but not always (Gowal et al., 2020) helpful for AT. Another branch of studies aims to improve the training efficiency of adversarial training based on PGD attack, termed as FAST AT (de Jorge et al., 2022; Jia et al., 2022b; Park & Lee, 2021; Wong et al., 2020; Andriushchenko & Flammarion, 2020; Jia et al., 2022a). Specifically, FGSM attack is adopted in Wong et al. (2020); Andriushchenko & Flammarion (2020); de Jorge et al. (2022) to replace PGD attack during training, which achieves promising robustness with catastrophic overfitting problem tackled.

**Dual BN in AT.** Prior work (Xie et al., 2020a) shows that adversarial samples can be used to improve recognition (accuracy) by adversarial training where adversarial samples are normalized by an independent $\text{BN}_{adv}$. Moreover, Xie & Yuille (2020) has shown that adding clean images in adversarial training (AT) can significantly decrease robustness performance, where such negative effects can be alleviated to a large extent by simply normalizing clean samples with an independent $\text{BN}_{clean}$. Inspired by their finding, Jiang et al. (2020) also adopts Dual BN in adversarial contrastive learning, showing that single BN performs significantly worse than Dual BN. Beyond Dual BN, triple BN has been attempted in Fan et al. (2021) for

incorporating another adversarial branch. Wang et al. (2021) has also combined Dual BN with Instance Normalization to form Dual batch-and-Instance Normalization for improving robustness. Prior work (Xie & Yuille, 2020) interprets the necessity of Dual BN from the perspective of an inherent large adversarial-clean domain gap. By contrast, Rebuffi et al. (2023) demonstrates that separate batch statistics are not necessary for Hybrid-AT and it is sufficient to use adapters with few domain-specific parameters for each type of input. Extensive experimental results in Rebuffi et al. (2023) show that the proposed model matches Dual BN's performance and the adversarial model soups perform better on ImageNet variants than the advanced masked auto-encoders. Our investigations extends the scope of Rebuffi et al. (2023) by reporting new findings on previously unaddressed topics, including but not limited to the role of disengaged NS in model training and the new understanding of adversarial-clean domain gap. Our two-task hypothesis extends the adapter method proposed in Rebuffi et al. (2023) by not only underpinning the adapter method with empirical evidence but also serving as a foundation for various methods utilizing domain-specific trainable parameters. This hypothesis also links the adapter method to broader Hybrid-AT enhancement strategies, such as Trades-AT (Zhang et al., 2019b). Additionally, the adapter's success in Rebuffi et al. (2023) reinforces our hypothesis, highlighting how our work complements and expands upon the contributions of Rebuffi et al. (2023).

**BN applications beyond AT.** Prior work investigates BN in various fields beyond AT, such as for adversarial transferability (Benz et al., 2021b; Dong et al., 2022). Benz et al. (2021b) investigates BN from the non-robust feature perspective. Specifically, Benz et al. (2021b) empirically reveals that BN shifts a model towards being more dependent on the non-robust features. Based on this finding, Benz et al. (2021b) suggests strategies like removing BN or early stopping during the training of substitute models to improve adversarial transferability. Another work Dong et al. (2022) provides both empirical and theoretical evidence which shows that the upper bound of adversarial transferability is influenced by the types and parameters of normalization layers. Based on this observation, (Dong et al., 2022) proposes a Random Normalization Aggregation (RNA) module to replace original normalization layers and create a combination of different sampled normalization. Extensive experiments demonstrate that the proposed RNA module achieves superior performance on different datasets and models. Another branch of work adopts Dual BN in domain adaptation. AdaBN (Li et al., 2017) leverages different statistics for two domains but loses source domain information by using only target domain statistics during inference. DSBN (Chang et al., 2019) introduces a separate BN branch for unsupervised domain adaptation, extendable to multisource scenarios. Huang et al. (2023) proposes reciprocal normalization that structurally aligns the source and target domains by conducting reciprocity across domains. In this work, we mainly focus on investigating Dual BN in the Hybrid-AT.

# 7 Conclusion

We experiment with Cross-AT and demonstrate the compatibility of clean samples' BN statistics with the adversarial branch, which inspires us to doubt the claims of prior work for justifying the necessity of Dual BN in Hybrid AT. We investigate the effect of disentangled NS and AP on training a Hybrid-AT model, leading to a thorough refutation of prior claims about the significance of NS. Our work further identifies a visualization flaw of the prior two-domain hypothesis, and points out that the adversarial-clean domain gap is not as large as expected. In addition, we propose a new interpretation of Hybrid-AT with Dual BN from the two-task perspective. Finally, we investigate different types of NS and AP in test-time, revealing that AP characterizes robustness during inference. Our study provides a comprehensive understanding of Dual BN as well as the adversarial examples.

**Acknowledgments**

This work was conducted by Center for Applied Research in Artificial Intelligence (CARAI) grant funded by DAPA and ADD (UD230017TD).

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

# A   Appendix

## A.1   Experimental Setups

**Experimental setups.** We train and evaluate all models in the paper using the same setups following Pang et al. (2020). In this work, we perform experiments on CIFAR10 (Krizhevsky et al., 2009; Andriushchenko & Flammarion, 2020; Zhang et al., 2022) with ResNet18 (Andriushchenko & Flammarion, 2020; Targ et al., 2016; Wu et al., 2019; Li et al., 2016; Zhang et al., 2022) and follow the suggested training setups in Pang et al. (2020) unless specified. Specifically, we train the model for 110 epochs. The learning rate is set to 0.1 and decays by a factor of 0.1 at the epoch 100 and 105. We adopt an SGD optimizer with weight decay $5 \times 10^{-4}$. For generating adversarial examples during training, we use $\ell_\infty$ PGD attack with 10 iterations and step size $\alpha = 2/255$. For the perturbation constraint, $\epsilon$ is set to $\ell_\infty$ $8/255$ (Pang et al., 2020) or $16/255$ (Xie & Yuille, 2020). Following Pang et al. (2020), we evaluate the model robustness under PGD-10 attack (PGD attack with 10 steps) and AutoAttack (AA) (Croce & Hein, 2020).

**Re-calibrate NS.** The $\mathrm{NS}_{clean}^{clean}$ and $\mathrm{NS}_{adv}^{adv}$ correspond to original Dual BN's $\mathrm{NS}_{clean}$ and $\mathrm{NS}_{adv}$, respectively. For re-calibrated NS, we simulate vanilla NS calculations, specifying the input domain following the subscript and AP following the superscript. For example, $\mathrm{NS}_{clean}^{adv}$ is computed by processing clean images through the branch with $\mathrm{AP}_{adv}$. To simulate the computation of running means in the original Dual BN, we forward the samples for multiple epochs for converged results. In Table 7, we first calculate various types of NS and then evaluate robustness by combining NS with AP.

**Training details of models in Section 5.3.** For the Dual Linear model, we add two linear layers above its penultimate layer and train it using the Hybrid loss outlined in Equation 1. For models with regularization loss, we train using a single BN and augment the original Hybrid-AT loss with an additional KL-divergence between the predictions on adversarial and clean inputs.

## A.2   Further investigations beyond BN

Inspired by finding that two sets of AP can achieve comparable results to Dual BN, we further investigate whether this holds in cases beyond BN where disentangling NS is not applicable. For example, layer normalization (LN) adopts sample-wise NS, and therefore it is not applicable to disentangle distribution-wise NS between two domains. We experiment with dual AP on ResNet with LN and the results are reported in Table 8. We observe that with Dual AP, LN performs similarly with BN in either setup (b) and (c) in Figure 3. We also investigate other normalization methods and model architectures (such as ViT) in Table 8. Table 8 shows that across various normalization methods and architectures, two AP sets achieve comparable performance to Dual BN while single AP set fails to achieves high robustness against AA.

Table 8: Effect of dual AP on various types of normalizations and models($\epsilon = 16/255$), where LN, GN and IN denote Layer Normalization, Group Normalization and Instance Normalization, respectively.

| Norm | Norm | Setups | Branch | Clean | PGD10 | AA |
|---|---|---|---|---|---|---|
| ResNet | BN | Single BN | / | 93.70 | 29.86 | 0.48 |
|  |  | Dual BN | $\mathrm{BN}_{adv}$ | 61.84 | 31.67 | 22.51 |
|  |  | Dual BN | $\mathrm{BN}_{clean}$ | 94.18 | 0.00 | 0.00 |
|  | LN | Single AP | / | 75.12 | 18.81 | 11.80 |
|  |  | Dual AP | $\mathrm{AP}_{adv}$ | 62.56 | 26.98 | 16.90 |
|  |  | Dual AP | $\mathrm{AP}_{clean}$ | 88.41 | 0.00 | 0.00 |
|  | GN | Single AP | / | 81.85 | 21.94 | 14.50 |
|  |  | Dual AP | $\mathrm{AP}_{adv}$ | 70.27 | 29.36 | 18.30 |
|  |  | Dual AP | $\mathrm{AP}_{clean}$ | 91.82 | 0.00 | 0.00 |
|  | IN | Single AP | / | 92.55 | 23.06 | 1.20 |
|  |  | Dual AP | $\mathrm{AP}_{adv}$ | 52.29 | 25.27 | 16.10 |
|  |  | Dual AP | $\mathrm{AP}_{clean}$ | 92.35 | 0.00 | 0.00 |
| ViT | LN | Single AP | / | 92.21 | 33.60 | 1.84 |
|  |  | Dual AP | $\mathrm{AP}_{clean}$ | 58.02 | 30.08 | 12.44 |
|  |  | Dual AP | $\mathrm{AP}_{adv}$ | 91.60 | 0.00 | 0.00 |

