# OpenReview forum: "Towards Understanding Dual BN In Hybrid Adversarial Training"
_TMLR — Accepted by TMLR_

### Review · Reviewer_eqrk · 2024-01-08

**Summary Of Contributions:**

This paper proposes a new perspective to understand Dual BN in Hybrid AT. Previous research has commonly supported the idea that there exists a significant domain disparity between adversarial and clean images. Moreover, prior studies have suggested that the implementation of Dual BN in Hybrid AT is effective due to disentangled statistics. However, this paper strongly challenges these assertions through extensive experimentation. One conclusion is that the key factors in Dual BN are two sets of affine parameters instead of disentangled statistics. Furthermore, the paper points out that the adversarial-clean domain gap is not as large as many might expect, and proposes a new interpretation of Hybrid-AT with Dual BN from the two-task perspective.

**Audience:**

Yes

**Broader Impact Concerns:**

None.

**Claims And Evidence:**

Yes

**Requested Changes:**

As mentioned in the weaknesses.

1. Please discuss other architectures, such as ViT models.
2. Please provide more details of different types of NS in Section 5.
3. Please discuss relevant papers.

**Strengths And Weaknesses:**

Strengths:

1. This paper is well-written and easy to follow.

2. This paper focuses on an important topic of applying Dual BN in Hybrid AT. Given the zero robustness of $BN_{clean}$ in original Dual BN, it is novel and impressive to find that $BN_{clean}$ can achieve robustness if it is trained with the adversarial branch.

3. This is the first paper to refute the two-domain hypothesis in prior studies. The authors point out that the adversarial-clean domain gap is not as large as expected. This finding is novel and strongly supported by solid experiments.

4. The proposed two-task hypothesis shows the possibility of alternative solutions beyond Dual BN to mitigate the adv-clean conflict in Hybrid AT. This update of understanding may inspire future studies on Hybrid AT from a new perspective.


Weaknesses:

1. Section 4 discusses the ResNet model with various normalizations such as LN. How about other architectures such as the ViT model?

2. It is unclear how different types of NS are computed in Section 5. More details are needed.

3. There exist several papers studying the role of normalization layers in adversarial robustness, which have not been discussed in this paper, such as [a, b].

[a]. Batch Normalization Increases Adversarial Vulnerability and Decreases Adversarial Transferability: A Non-Robust Feature Perspective. ICCV 2021.

[b]. Random Normalization Aggregation for Adversarial Defense. NeurIPS 2022.

---

> ### Author Response · Authors · 2024-02-08
> **Thank you for your valuable reviews. We address the questions below.**
>
> Thank you for your valuable reviews. We are glad that you find our work to focus on an important topic of applying Dual BN in Hybrid AT and refute the two-domain hypothesis in prior studies, and the update of understanding may inspire future studies on Hybrid AT from a new perspective. We address your questions below.
>
>
> > ### **1. Results on ViT model**
>
> Thank you for your suggestions. We report the results of ViT models in the following Table 1. The results show that disentangled AP in ViT model can achieve comparable results to Dual BN, while a single AP set fails to achieve good robustness. This trend is same with the ResNet model. We add the results of ViT model in the appendix (Section A.2) of the revised paper.
>
> Table 1: Results of disentangling AP in ViT model
>
> | Model         | Norm   | Branch   | Clean | PGD Acc | AA    |
> |---------------|----------|----------|-------|---------|-------|
> | ResNet        | Single BN | /      | 93.70 | 29.86   | 0.48  |
> |               | Dual BN |$BN_{clean}$ | 61.84 | 31.67   | 23.14 |
> |               | Dual BN | $BN_{adv}$ | 94.18 | 0.00    | 0.00  |
> | ResNet        | Single LN |/      | 75.12 | 18.81   | 11.80  |
> |               | Dual LN |$LN_{clean}$ | 62.56 | 26.98   | 16.90 |
> |               | Dual LN|  $LN_{adv}$ | 88.41 | 0.00    | 0.00  |
> | ViT     | Single LN   |/    | 92.21 | 33.60   | 1.84  |
> |         | Dual LN  | $LN_{clean}$   | 58.02 | 30.08   | 12.44 |
> |         | Dual LN  |$LN_{adv}$  | 91.60 | 0.00    | 0.00  |
>
>
>
> > ### **2. Details in the computation of different NS.**
>
> Thank you for your suggestions. We have added detailed setups in the appendix (Section A.1) of the revised paper.
>
> The $NS_{clean}^{clean}$ and $NS_{adv}^{adv}$ correspond to original Dual BN's $NS_{clean}$ and $NS_{adv}$, respectively. For re-calibrated NS, we simulate vanilla NS calculations, specifying the input domain following the subscript and AP following the superscript. For example, $NS_{clean}^{adv}$ is computed by processing clean images through the branch with $AP_{adv}$. To simulate the computation of running means in the original Dual BN, we forward the samples for multiple epochs for converged results.
>
>
>
>
> > ### **3. Add discussions on related papers.**
>
>
> Thank you for suggesting related papers[1,2]. We have cited and discussed them in Section 6 of the revised paper.
>
> [1] investigates  BN from the non-robust feature perspective. Specifically, [1] empirically reveals that  BN shifts a model towards being more dependent on the non-robust features. Based on this finding,  [1] suggests strategies like removing BN or early stopping during training of substitute models to improve adversarial transferability.  Another work [2] provides both empirical and theoretical evidence which shows that  the upper bound of adversarial
> transferability is influenced by the types and parameters of normalization layers. Based on this observation, [2] proposes a Random Normalization Aggregation (RNA) module to replace original normalization layers and create a combination of different sampled normalization. Extensive experiments demonstrate that the proposed RNA module achieves superior performance on different datasets and models.
>
> [1]. Batch Normalization Increases Adversarial Vulnerability and Decreases Adversarial Transferability: A Non-Robust Feature Perspective. ICCV 2021.
> [2]. Random Normalization Aggregation for Adversarial Defense. NeurIPS 2022.

---

### Review · Reviewer_Dqef · 2024-01-21

**Summary Of Contributions:**

This work investigates Dual BN in hybrid adversarial training (AT) with extensive experiments. As Hybrid AT model takes both adversarial and clean images as inputs, it is popular to adopt Dual BN for higher performance. According to prior two-domain hypothesis, Dual BN’s effectiveness is attributed to its disentangled normalization statistics. By contrast, this work finds that the disentangled affine parameters play the most important role in Dual BN instead of the statistics. This paper also refutes prior two-domain hypothesis by pointing out a visualization flaw. It further introduces the two-task hypothesis that demonstrates the potential of other methods. The topics and findings are of interest to the community.

**Audience:**

Yes

**Claims And Evidence:**

Yes

**Requested Changes:**

See the weaknesses part:
* Provide more discussions of applying Dual BN in other applications rather than adversarial training.
* Provide more training details for section 5.3 and more detailed experimental setups for Table 5.

**Strengths And Weaknesses:**

Strengths

* This work provides multiple novel and interesting findings on Dual BN. For example, it is novel and important to prove that disentangled affine parameters play the main role in Dual BN instead of the normalization statistics claimed in prior work.
* This paper points out a clue to refute the previous two-domain hypothesis, and proposes the new two-task hypothesis. Following the two-task hypothesis, multiple methods can achieve comparable performance to Dual BN. This finding is novel and may inspire more studies in the future.
*  The analysis and experiments are solid to validate the conclusions. The paper is well organized.

Weaknesses

* In related works, this paper only discusses how to apply Dual BN in adversarial training. Discussions of applying Dual BN in other applications are needed.
* Section 5.3 includes multiple models such as Dual Linear model and model with regularization loss. How are these models trained?
* The experimental setups in Table 5 are unclear.

---

> ### Author Response · Authors · 2024-02-08
> **Thank you for your valuable reviews. We address the questions below.**
>
> Thank you for your valuable reviews. We are glad that you find our work provides multiple novel and interesting findings on Dual BN, points out a clue to refute the previous two-domain hypothesis, and also includes solid analysis and experiments to validate the conclusions. We address your questions below.
>
> > ### **1. Discussions on Dual BN beyond adversarial training**
>
>
> Thank you for your suggestions. In response, we added discussions on Dual BN beyond adversarial training in the revised paper, such as domain adapation. Please refer to the purple texts in Section 6.
>
>
> > ### **2. Training setups of models in Section 5.3**
>
>
>
> Thank you for your comments. We summarize the training setups of models in Section 5.3 as follows, which are included in the appendix  (Section A.1) of the revised paper.
>
> For the models discussed in Section 5.3, we use the same training setups as the vanilla Hybrid-AT models in the paper. Following the suggested setups in [1],  we train each model for 110 epochs with an initial learning rate of 0.1, which decays by a factor of 0.1 at epochs 100 and 105. We use an SGD optimizer with a weight decay of $5\times10^{-4}$. Adversarial examples are generated with an $\ell_{\infty}$ PGD attack, using 10 iterations and a step size of $\alpha=\epsilon/4$. For the Dual Linear model, we add two linear layers above its penultimate layer and train it using the Hybrid loss outlined in the paper's Equation 1. For models with regularization loss, we train it using a single BN and augment the original Hybrid-AT loss with an additional KL-divergence between the predictions on adversarial and clean inputs.
>
> [1] Tianyu Pang, Xiao Yang, Yinpeng Dong, Hang Su, and Jun Zhu. Bag of tricks for adversarial training. ICLR, 2021
>
> > ### **3. Experimental setups of Table 5**
>
> Thank you for your suggestions. We have added detailed setups of Table 5 in the appendix (Section A.1) of the revised paper.
>
> In Table 5, we first calculate various types of NS and then evaluate robustness by combining NS with AP. The $NS_{clean}^{clean}$ and $NS_{adv}^{adv}$ correspond to original Dual BN's $NS_{clean}$ and $NS_{adv}$, respectively. For re-calibrated NS, we simulate vanilla NS calculations, specifying the input domain following the subscript and AP following the superscript. For example, $NS_{clean}^{adv}$ is computed by processing clean images through the branch with $AP_{adv}$. To simulate the computation of running means in the original Dual BN, we forward the samples for multiple epochs for converged results.

---

### Review · Reviewer_iNQL · 2024-01-28

**Summary Of Contributions:**

The paper investigates the reasons for the success of dual batch normalization (Dual BN) in adversarial training, i.e., using different batch normalization layers to process clean and adversarial samples. In particular, the authors debunk the previous hypothesis that Dual BN works because it mitigates the distribution shift introduced by the adversarial perturbations during training, and show, instead, that Dual BN works because it introduces parallel sets of trainable parameters that can be used to independently learn to solve the clean and adversarial tasks. The claims of the authors are verified using different empirical ablation studies that look at different architecture variants of batch normalization (and other normalization layers) and by inspecting the activation statistics of a network under different settings.

**Audience:**

Yes

**Broader Impact Concerns:**

I don't see any major societal concerns stemming from this work.

**Claims And Evidence:**

Yes

**Requested Changes:**

Due to the major overlap with prior work I cannot unfortunately recommend the acceptance of this manuscript and will not be requesting any changes over it.

**Strengths And Weaknesses:**

**Strengths**

1. **Clear message**: The paper has a solid and clean message which is conveyed very clearly. The problem under study is very technical and niche, but could still be of broad interest to people working on adversarial robustness.
2. **Strong empirical verification**: I find that the paper follows a very good scientific methodology to verify all its claims. The ablation studies decoupling the effect of normalization statistics and trainable parameters in Dual BN are very clear and solid.

**Weaknesses**
1. **MAJOR: Same results were previously published in ICLR2023**: Unfortunately, the exact same observation and explanation of Dual BN has already been published by [Croce et al. (2023)](https://openreview.net/pdf?id=HPdxC1THU8T). Although this paper uses different experiments to arrive at the same conclusion, it conveys exactly the same message, and goes, in fact, beyond this observation to construct a new method to interpolate between adversarial and clean behaviors during inference by mixing the classification token of an adversarially trained ViT. I invite the authors to read that work (specifically Section 3.2) and let me know if they think their work provides something else on top of Croce et al. (2023), but at the moment I am very inclined to reject this work on the basis of lack of novelty.
2. **Minor: Overly repetitive text**: I find the first few pages of the paper a bit repetitive. The dichotomy between the old hypothesis of Xie & Yuille (2020) and the one of this work is repeated too much in my opinion. The writing could be streamlined and go more directly to the experimental validations. Overall, however, I find the paper is well written and easy to follow.

---

> ### Author Response · Authors · 2024-02-08
> **Thank you for your valuable reviews. We address your questions below. (Response 1/3)**
>
> Thank you for your valuable reviews. We are glad that you find our work clearly conveys a solid and clean message, and presents strong empirical verification. We address your questions below.
>
> > ### **1. Differences between our work and Croce et al[1]**
>
> Thank you for suggesting the related paper Croce et al[1]. We acknowledge that our work intersects with [1] in reporting that disentangled affine parameters can match Dual BN's performance during training. However, we clarify that our work differs from [1] and provides **four new findings** regarding Dual BN on top of [1].
>
> We summarize the differences between our work and [1], as well as our unique findings in the following. We also revised the whole paper to emphasize our contributions. We cite and discuss [1] in Section 1, Section 4, Section 5, and Section 6 of the revised paper. Please refer to the blue texts in the revised paper for detailed discussions.
>
> [1] Sylvestre-Alvise Rebuffi, Francesco Croce, and Sven Gowal. Revisiting adapters with adversarial training. ICLR, 2023.
>
> > ### **1.1. Summary of differences from Croce et al[1]**
>
> * Despite the insightful investigations in [1], our research identifies key questions that **remain unanswered**, highlighting the necessity of our work for further exploration of Dual BN. (Please refer to the following 1.2.)
>
> * Our research offers a detailed exploration of Dual BN, covering topics **not addressed in [1]**, including but not limited to understanding the adversarial-clean domain gap. Consequently, our study **uniquely reveals four critical findings absent from [1]**. (Please refer to the following 1.3.)
>
> * Our two-task hypothesis extends the adapter method proposed in [1] by not only underpinning the adapter method with empirical evidence but also serving as a foundation for various methods utilizing domain-specific trainable parameters. This hypothesis also links the adapter method to broader Hybrid-AT enhancement strategies, such as Trades-AT. Additionally, the adapter's success in [1] reinforces our hypothesis, highlighting how our work complements and expands upon the contributions of [1]. (Please refer to the following 1.4.)
>
>
> > ### **1.2: Questions unaddressed by [1]**
>
> * Question 1: Can disentangled normalization statistics (NS) with a single set of affine parameters (AP) match Dual BN's robustness?
>
> * Question 2: Why is the NS gap between clean and adversarial domains so significant, as highlighted in the original Dual BN paper[2]?
>
> * Question 3: Expanding upon [1]'s exploration of Dual BN during model training, how does Dual BN work in test time? For example, during evaluation,  does replace $NS_{adv}$ with $NS_{clean}$ in original $BN_{adv}$ yield comparable robustness?
>
> [2] Cihang Xie and Alan Yuille. Intriguing properties of adversarial training at scale. ICLR, 2020.

---

> ### Author Response · Authors · 2024-02-08
> **Continue (Response 2/3)**
>
> > ### **1.3: Unique findings in our work and answers to the above questions**
>
> > **1.3.1: Finding 1: Disentangled NS matches Dual BN's robustness under specific conditions（Section 4）**
>
> **Experiment overview:**  Our study extends beyond [1] by not only comparing models with disentangled AP sets but also examining the impact of mixed versus disentangled NS. The results are reported in Setup2 of Table 3.
>
> **Unique finding:**  We find that disentangled NS alone can match Dual BN's robustness under a small perturbation ($\epsilon=8/255$), even against AutoAttack. However, with larger perturbations ($\epsilon=16/255$), its advantage is much smaller. We report this finding in Section 4.
>
> **Contributions and differences from [1]:** While [1] explores disentangled affine parameters (AP), it overlooks the investigation of disentangled normalization statistics (NS), leaving a gap in the literature. Our study addresses this by comparing the effects of disentangling both AP and NS, thereby offering a thorough understanding of Dual BN and answering Question 1.
>
>
> > **1.3.2: Finding 2: Identifying a visualization flaw of the large adversarial-clean domain gap in the original Dual BN paper[2]. （Section 5.1）**
>
> **Experiment overview:** We are the first to highlight that NS calculation depends on both input images and AP, e.g., original $NS_{clean}$ is derived from clean images and $AP_{clean}$.  We introduce re-calibrated NS for comparison using mismatched inputs and APs, such as $NS_{clean}^{adv}$ from clean images with $AP_{adv}$, illustrated in Figure 4.
>
> **Unique finding:** We identify a visualization flaw of the large adversarial-clean domain gap in [2], attributed to AP inconsistency. Specifically, $NS_{clean}$ and $NS_{adv}$ are calculated with $AP_{clean}$ and $AP_{adv}$, respectively. Using the same AP for NS calculation markedly reduces this gap, as detailed in Section 5.1.
>
> **Contributions and differences from [1]:** Acknowledging the findings of 'not necessary NS' in [1] and the reported large domain gap in [2] may seem contradictory, creating potential confusion. Unlike [1], which does not explain the visualized domain gap in [2], our work identifies a visualization flaw in [2], addressing and clarifying Question 2. This clarification is both timely and essential toward a full understanding on the adv-clean domain gap as well as Dual BN.
>
> > **1.3.3: Finding 3:  Adversarial-clean domain gap is not as large as expected. （Section 5.2）**
>
> **Experiment overview:** Building on Finding 2's visual insights, we further quantitatively evaluate the adversarial-clean domain gap with two key experiments:
> * We measure the NS gap and AP gap between clean and adversarial images using Wasserstein distance, shown in Figure 6.
> * We compare the adversarial-clean domain gap with the noisy-clean domain gap, detailed in Table 6 and Figure 7.
>
> **Unique finding:** The above experiments, together with the visual findings in Finding 2, show that the adversarial-clean domain gap is not as large as expected and is comparable to the noisy-clean gap.
>
> **Contributions and differences from [1]:** While [1] examines the impact of NS in training robust models, it does not explore the adversarial-clean domain gap. Our integration of Finding 2 and Finding 3 significantly enriches the understanding of the adversarial-clean domain gap and the nature of adversarial examples.
>
> > **1.3.4: Finding 4: AP characterizes robustness in test-time（Section 5.4）**
>
> **Experiment overview:**  Using a trained model, we evaluated the robustness of different NS and AP pairs, detailed in Table 7.
>
> **Unique finding:** We find that AP characterizes the large robustness gap between $BN_{clean}$ and $BN_{adv}$ during inference.  Replacing $NS_{adv}$ with $NS_{clean}$ in original $BN_{adv}$ yields an above zero robustness but is still lower than original $BN_{adv}$. However, when AP is consistent for both NS computation and robustness evaluation, the robustness gap between the NS calculated on clean or adversarial samples is limited.
>
> **Contributions and differences from [1]:** Building upon [1]'s investigation of Dual BN in model training, our findings provide a new insight by investigating BN in test-time, offering a comprehensive understanding of Dual BN.

---

> ### Author Response · Authors · 2024-02-08
> **Continue (Response 3/3)**
>
> > ### **1.4: Distinguishing our work from the adapter method in [1]. （Section 5.3 and Section 6）**
>
> * **New empirical foundation for [1].**  Our work builds on [1]'s contributions, which use adapters based on the correlation between two sets of AP and the adapter literature, highlighting their role as domain-specific trainable parameters. We advance this with our two-task hypothesis for Hybrid-AT, extending the adapter method from [1]. This hypothesis not only underpins the adapter method with empirical evidence but also serves as a foundation for various methods utilizing domain-specific trainable parameters.
>
> * **A unified framework.** Our two-task hypothesis extends [1] by linking the adapter method to broader Hybrid-AT enhancement strategies, such as Trades-AT.  This unified framework offers new insights for future research in Hybrid-AT.
>
> * **Adapters in [1] validate our work**  The superior performance of the adapter method in [1] also strongly validates the effectiveness of our two-task hypothesis, highlighting how our work complements and expands upon the contributions of [1].
>
> > ### **2. More concise writing**
>
> Thank you for your insightful suggestions. In response, we have revised the paper to streamline repetitive content on the differences between our work and [2], particularly in Sections 1, 2, 4, and 5.  Please refer to the blue texts in the revised paper.

---

> > ### Comment · Reviewer_iNQL · 2024-02-08
> > **Reply to authors**
> >
> > I sincerely thank the authors for seriously considering my comments and for their very honest effort at placing their work within the right context in the literature. The new version of the manuscript very generously cites Rebuffi et al (2023) and gives the right credit to their ideas. I also acknowledge the efforts of the authors in describing how their works complements Rebuffi et al (2023) which has been clearly addressed in their rebuttal and the manuscript revision. Personally, I believe the new findings of this work are not groundbreaking or deeply transformative, but because they are grounded in good empirical science and are properly contextualized with respect to the literature I am now leaning towards acceptance. As I see it, the main contribution of this work is a fresh replication of the findings of Rebuffi et al (2023), including new ablations and a discussion from a different angle. There may be few people interested in this in the community and so I believe the work can be accepted.
> >
> > The new text is more streamlined and clear to read, and I thank the authors for their effort. However, to not mislead the community when browsing this paper, I would suggest that a citation to Rebuffi et al (2023) is also included in the abstract acknowledging that this paper is an extension of their findings.

---

> > > ### Author Response · Authors · 2024-02-08
> > > **Thank you for your insightful feedback**
> > >
> > > Thank you for your insightful feedback. We thank you for your time in evaluating this work and find the significant improvements in our revised manuscript. Following your suggestions, we have now cited and discussed Rebuffi et al (2023)[1] in the abstract of our updated paper.
> > >
> > > [1] Sylvestre-Alvise Rebuffi, Francesco Croce, and Sven Gowal. Revisiting adapters with adversarial training. ICLR, 2023.

---

### Decision · Action_Editor_Zcko · 2024-03-07

**Recommendation:** Accept as is

**Comment:**

This paper introduces a novel dual batch normalization method tailored for adversarial training. The reviewers unanimously commend the manuscript for its significant contribution, highlighting intriguing insights into Dual BN, robust empirical validation, and lucid hypotheses. Minor concerns raised during the rebuttal phase have been effectively addressed through additional results and analysis. Furthermore, the consensus among the reviewers is that this paper will captivate readers, garnering positive feedback.

Overall, this manuscript fits the TMLR acceptance criteria and I recommend accepting this paper.

**Audience:**

Yes.

**Claims And Evidence:**

Yes. The claims made in the submission are well supported by analysis and experiments.